# Modification of Structural Properties Using Process Parameters and Surface Treatment of Monolithic and Thin-Walled Parts Obtained by Selective Laser Melting

**DOI:** 10.3390/ma13245662

**Published:** 2020-12-11

**Authors:** Krzysztof Grzelak, Janusz Kluczyński, Ireneusz Szachogłuchowicz, Jakub Łuszczek, Lucjan Śnieżek, Janusz Torzewski

**Affiliations:** Institute of Robots & Machine Design, Faculty of Mechanical Engineering, Military University of Technology, 2 Gen. S. Kaliskiego St., 00-908 Warsaw, Poland; krzysztof.grzelak@wat.edu.pl (K.G.); ireneusz.szachogluchowicz@wat.edu.pl (I.S.); jakub.luszczek@wat.edu.pl (J.Ł.); lucjan.sniezek@wat.edu.pl (L.Ś.); janusz.torzewski@wat.edu.pl (J.T.)

**Keywords:** additive manufacturing, selective laser melting, microstructural investigation, porosity, surface treatment, 316L stainless steel

## Abstract

Additive manufacturing is one of the most popular technological processes and is being considered in many research works, a lot of which are related to thin-walled parts analysis. There are many cases where different part geometries were manufactured using the same process parameters. That kind of approach often causes different porosity and surface roughness values in the geometry of each produced part. In this work, the porosity of thin-walled and monolithic parts was compared. To analyze additively manufactured samples, porosity and microstructural analyses were done. Additionally, to check the influence of process parameter modification on the manufactured parts’ properties, hardness and roughness measurements were made. Surface roughness and the influence of surface treatment were also taken into account. Porosity reduction of thin-walled parts with energy density growth was observed. Additionally, a positive influence of slight energy density growth on the surface roughness of produced parts was registered. Comparing two extreme-parameter groups, it was observed that a 56% energy density increase caused an almost 85% decrease in porosity and a 45% increase in surface roughness. Additional surface treatment of the material allowed for a 70–90% roughness reduction.

## 1. Introduction

Additive manufacturing is currently one of the most popular topics of different research papers [1]. There are many publications connected with different types of analysis. Among these, the authors consider material properties, the numerical analysis of additively manufactured parts, and case studies of specific parts. Laser powder bed fusion (L-PBF) technologies are characterized by using an energy source to melt metallic powder grains [2] where the volume of the material is constituted on melt pools which create a specific, layered structure [3,4]. Additively manufactured parts were used for the authors’ own research connected with static, dynamic, and fatigue behavior [5,6,7,8] with additional process modification [9] and heat treatment [10]. Regardless of the process parameters selected, different material behavior was observed during the testing of monolithic parts and thin-walled or lattice-structured elements. The mentioned phenomenon was observed by Sienkiewicz et al. [2], where geometrical accuracies and microstructures revealed some weaknesses of the selected manufacturing process, such as the deviation of the dimensions of lattice struts measured on the frontal plane. Additionally, the authors of the mentioned work revealed the presence of imperfections such as porosity, voids, and unmelted powder grains. These kinds of issues must be taken into account during finite element analysis (FEA).

One of the most popular approaches is using the plastic strain value adopted in FEA; a good example was used by Kucewicz et al. [11], where the authors implemented a meshless method of modeling the orthotropic properties of the material. This method represents a kind of compromise between microscopic and macroscopic levels. In other research papers [12], it was stated that increasing the value of relative density causes a growing sensitivity of the structure to strain rate effects during the testing of thin-walled honeycomb structures made of Ti6Al4V alloy. In some research papers, it was observed that the honeycomb structures are characterized by brittle fracture in dynamic tests, which resulted in significantly worse energy-consuming parameters for that kind of structure [13]. Different material behavior after the L-PBF processing of thin-walled parts has to be taken into account by designers, especially when constructing parts dedicated to military, aircraft, and medical applications [14,15,16,17,18,19].

Design assumptions dedicated to selectively laser melted (SLM) parts state that it is common to design self-supporting structures to minimize the use of support structures [20,21]. In many cases, the use of bridge structures, which are not self-supporting, is recommended. That type of structure is used two connect points without any support structure from below. These kinds of short bridges can be obtained without any support structures, and it is assumed they will be used to save material and manufacturing process time.

Geometrical complexity significantly affects heat transfer into the material volume during the manufacturing process. The phenomenon of heat transfer is strictly related to molten pool behavior during the L-PBF process, which in turns affects void generation. It has been described by Gusarov et al. [22], where authors point to transfer of the laser radiation in powder as being responsible for the production of volumetric heat. Created in this way, a melt pool’s flow is driven by its surface tension. Movement of the melt pool along the exposure line allows for its contact with the substrate in its central part, and with loose powder particles in its lateral part (where surface energy is reduced).

On the other hand, Liu et al. [23] described the influence of powder particles on heat transmission, where the local overheating of powder particles elicits material evaporation and generates recoil pressure on the molten pool. Further analysis of material behavior during L-PBF was performed by Matthews et al. [24], where the authors analyzed melt pool dynamics and vapor flow using the Lagrangian-motion component; they determined it was responsible for the adiabatic movement of the material in response to forces present in the considered volume of the melt pool.

During our preliminary research connected with the numerical analysis of honeycomb structures (not yet published), it was observed that there were significant differences between experimental results and numerical analysis. The main issue was connected with significant porosity growth in the area of the thin wall visible in Figure 1.

The presence of significant differences between the properties of monolithic and thin-walled parts obtained using the same material and additive manufacturing technology is considered in this paper. It determines how the manufacturing process of thin-walled parts affects the microstructure and porosity of that type of element. Additionally, five different process parameter groups were tested to analyze how to minimize the negative influence of manufacturing thin-walled structures on parts’ structural properties. The influence of the parameters used for thin-walled structures on their surface roughness was also taken into account.

## 2. Materials and Methods

### 2.1. Material

The material used for sample manufacturing was gas-atomized 316L stainless steel powder characterized by a spherical shape supplied by SLM Solutions AG company (SLM Solutions, Lubeck, Germany) with the grain size in the range of 15–45 µm and a flowability of 14.6 s/50 g. The chemical composition of the material is shown in Table 1.

The material properties of SLM-processed 316L steel (based on SLM Solutions’ Data Sheet) using 50 μm layer thickness are shown in Table 2.

The powder particles’ size distribution cumulated mass had values as follows: D10 = 18.22 μm, D50 = 30.50 μm, D90 = 55.87 μm. Scanning electron microscopy (SEM) images of the powder grains are shown in Figure 2.

### 2.2. Additive Manufacturing Process

A selective laser melting process was performed using an SLM 280HL machine (SLM Solutions, Lubeck, Germany) in an argon atmosphere. For sample manufacturing, five process parameters were used and are shown in Table 2. Abbreviations used in Table 3: L_p_—laser power, ev—exposure velocity, h_d_—hatching distance, l_t_—layer thickness, ρ_E_—energy density. Energy density is the main process parameter that creates a relationship between laser power, exposure velocity, hatching distance, and layer thickness.

These parameters were selected based on previous research experience [25] where significant differences during hatching distance changes were observed. Additive manufacturing of thin-walled structures is characterized by a higher cooling rate than occurs in monolithic parts, so it is very important to keep stable and slight change which can be controlled to a certain extent. That kind of characteristic was registered when changing laser power. For this research, five parameter groups were used and only laser power was changed to reach a stepwise movement in energy density equal to 10 J/mm^3^. That approach allowed the observation of material behavior after using different values of laser power.

### 2.3. Material Properties Analysis

Powder particles analyses were performed using a Jeol JSM-6610 (Jeol, Tokyo, Japan). The evaluation of porosity, microstructural observations, and roughness analysis using 3D laser measurement was performed using an Olympus LEXT 4100 confocal microscope (Olympus Corporation, Tokyo, Japan,). For porosity analysis, MountainsMap 6 software (version 7.2, Digital Surf, Besançon, France) was used. To reveal the microstructure, an “acetic glyceregia” solution heated to 40 °C was used as the etchant (6 mL HCl, 4 mL HNO_3_, 4 mL CH_3_COOH, and 0.2 mL glycerol). The etching time was 5 s.

For each manufactured part, Vickers hardness distribution measurements were taken using a Struers DuraScan 70 (Struers, Copenhagen, Denmark) hardness tester. All measurements were conducted in the monolithic part of the samples (shown in Figure 3). The samples’ geometry was designed in this way in order to analyze porosity in the monolithic and thin-walled parts and to check the material condition in the areas of the “bridges”. Their dimensions were as follows: 100 mm length, 10 mm width, and 12 mm height. Thin-walled parts (area “A” in Figure 3) were characterized by a thickness of 0.75 mm and 1 mm spacing. The monolithic part of the samples in the middle (area “B” in Figure 3) had dimensions of: 10 mm length, 10 mm width, and 12 mm height.

To reduce surface roughness, a PK 1200 E sandblaster with silica sand (Virgo, Siedlce, Poland) was also used. For the additional surface treatment of the additively manufactured (AM) parts, a Struers LectroPol (Struers, Copenhagen, Denmark) was used for electropolishing.

For surface-treated samples, each surface was electropolished three times in A3 Struers electrolyte using 35 V voltage, 5 A amperage, and 40 s polishing time.

## 3. Results and Discussion

### 3.1. Porosity Analysis

To compare the obtained results of porosity analyses, all registered values have been put in Table 4. To the extent possible, for each porosity result an area of the analyzed thin-walled structure has been attached.

Based on the obtained porosity results, an increased number of voids in samples manufactured using lower energy density could be observed. Indeed, a value of porosity lower than 1% is acceptable [26,27], but, as could be observed in Figure 1, using more complex thin-walled structures could affect porosity growth. Additionally, in the attached images in Table 2, the highest number of pores is visible in the middle of the thin-walled part of the sample, which could be affected by the varied thermal history of that volume of material in comparison to the monolithic part of the material. The recommended energy density (by SLM Solutions) for monolithic parts made of 316L steel is about 55 J/mm^3^. Considering the porosity results, from a porosity-reduction point of view, it is worthwhile to increase the energy density by about 30% [28] in order to assure appropriate material densification, especially during the manufacturing of thin-walled structures [12,13].

### 3.2. Microstructural Analysis

To extend material behavior analysis, microstructural analysis was conducted. The area of connection between the thin-walled and monolithic parts of the sample was taken into account, and is shown in Figure 4.

Considering the obtained microstructure, there is a visible non-homogenous and irregular structure of the material in the area of the bridge (area C in Figure 4), and in the connection between the outline shell (area E in Figure 4) and the core of the part (area F in Figure 4), some pores (D) are visible. The mechanism of porosity generation could be connected with spattering during the exposure of the outline shell, which could affect porosity generation in the core of the part, or the border between the outline and core part. Using a higher energy density effectively reduced porosity. Despite using different energy density values, the irregular structure of the bridges did not change, which from the fatigue properties perspective is a very negative phenomenon.

### 3.3. Surface Roughness Analysis

After observing a lack of visible improvement of the irregular shape in the area of the bridge of the manufactured samples, an influence of the used process parameters on surface roughness was determined. 

It has already been discovered that using different values of energy density affect surface roughness [29]. As has already been well-established, pores in the material structure could be responsible for fatigue cracking propagation [30,31,32]. The other issue which must be taken into account from a fatigue properties point of view is surface roughness. To justify parameter modification for porosity reduction, a surface roughness measurement was made. The obtained results are shown in Table 5.

Based on the obtained results, an increased value of average roughness surface profile (R_a_) in monolithic samples manufactured using a higher value of energy density could be observed. The highest value of the lowest and highest points distribution (shown as the same color—the red top of the first parameter set) in samples manufactured using the low energy density is also visible. The issue of slight increase and surface roughness stabilization near a value of 19 μm could be connected with the increased spattering caused by providing a high amount of energy to the material volume [33]. That phenomenon should be taken into consideration during process preparation, where it is necessary to distinguish part porosity, surface roughness, or both to reach desirable fatigue properties.

### 3.4. Hardness Analysis

For additional analysis, hardness measurements of the monolithic part of the samples were taken into account. The obtained results are shown in Figure 5.

Based on the obtained hardness measurements, an initial increase and then a slight decrease in hardness is visible; this was also observed in other research [7,10] and could be connected with a kind of heat treatment effect caused by a higher temperature delivered to the material volume. As was observed in [7], higher values of average hardness are caused by increased hardness in the layer connection zone.

All registered differences between the used process parameters and their influence on parts’ mechanical properties indicate better results when using a higher energy density. That phenomenon is positive to some limit, which is defined by the process technological window for each material. Using an energy density above the mentioned limit causes porosity growth connected with the “keyhole effect”, where porosity is generated by an unstable melt pool during SLM processing.

### 3.5. Surface Treatment

One of the most characteristic features of AM technology is the ability to obtain manufactured parts with complex geometry. This approach has a very significant influence on fatigue properties, especially when there is also a high surface roughness. As it was shown in Section 3.3, it is possible to reduce surface roughness during the SLM processing, but there are also surface treatment methods which seem to be appropriate for geometrically complex parts.

SLM-processed parts are full-metallic parts which could be subjected to different surface treatments available for metals. The most efficient surface treatments are methods using the kinetic energy of accelerated small parts (e.g., metal shooting or sanding) or different types of chemical and electrochemical polishing [26,34].

Structure images after different surface treatment operations with roughness distribution compared to as-built samples are shown in Table 6.

The surface treatments allowed for 71% roughness reduction after sanding and 91% reduction after electropolishing. The most favorable method from the geometrical complexity point of view is electropolishing, because of the ability of electrolytes to penetrate the entire area of the surface being treated. Based on the conducted tests, it is very important to ensure a similar roughness value across the part. Samples that were characterized by increased roughness in some areas had an unstable polishing process, which caused high structural heterogeneity of the surface. Such heterogeneity was characterized by increased roughness, and even reduced geometrical dimensions in those areas.

## 4. Conclusions

Understanding the influence of different process parameters in SLM additive manufacturing on the material microstructure and mechanical properties is very helpful during the design of special-purpose constructions. Based on the results obtained in this study, the following conclusions could be drawn:An energy density increase of 56% caused almost an 85% decrease in porosity and a 45% increase in surface roughness. Additionally, manufacturing using parameters with lower energy density led to a slight increase in the measured hardness.Using higher energy density during the SLM process reduced the layered structure effect and slightly increased surface roughness with the growth of the material solidification.For thin-walled structures, using a higher amount of energy density than would be used for monolithic parts is suggested to obtain better structural properties. This kind of modification has to be limited by process technological windows specific to each material.Surface treatment allowed for a 70–90% roughness reduction. In the case of electropolishing, it was very important to ensure a stable surface roughness value before the treatment in order to avoid structural heterogeneity generation after polishing.

## Figures and Tables

**Figure 1 materials-13-05662-f001:**
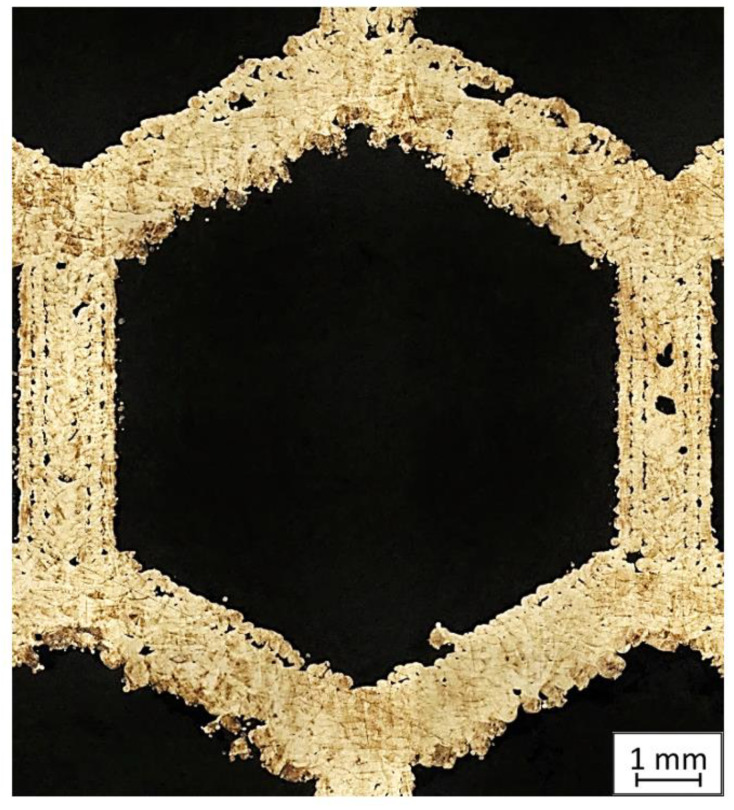
Honeycomb cell with visible porosity in specific areas.

**Figure 2 materials-13-05662-f002:**
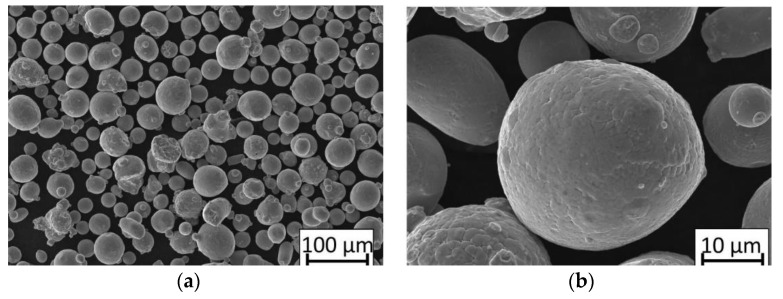
SEM images of 316L powder grains captured at scales of (**a**) 100 µm and (**b**) 10 µm.

**Figure 3 materials-13-05662-f003:**
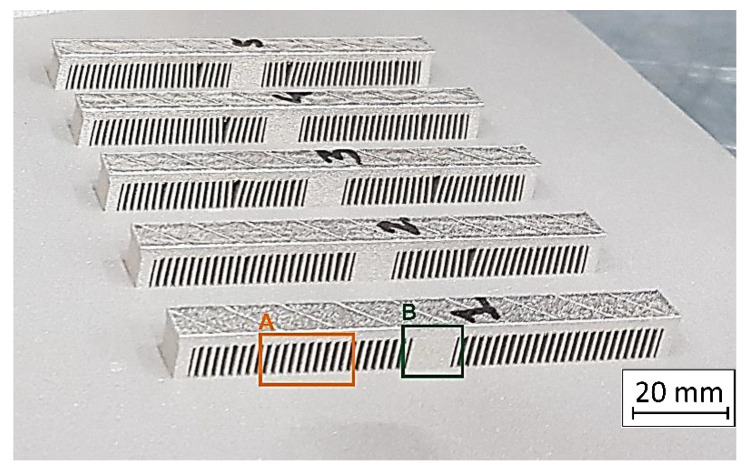
Samples for structural analysis with thin-walled areas (**A**) and a monolithic part (**B**).

**Figure 4 materials-13-05662-f004:**
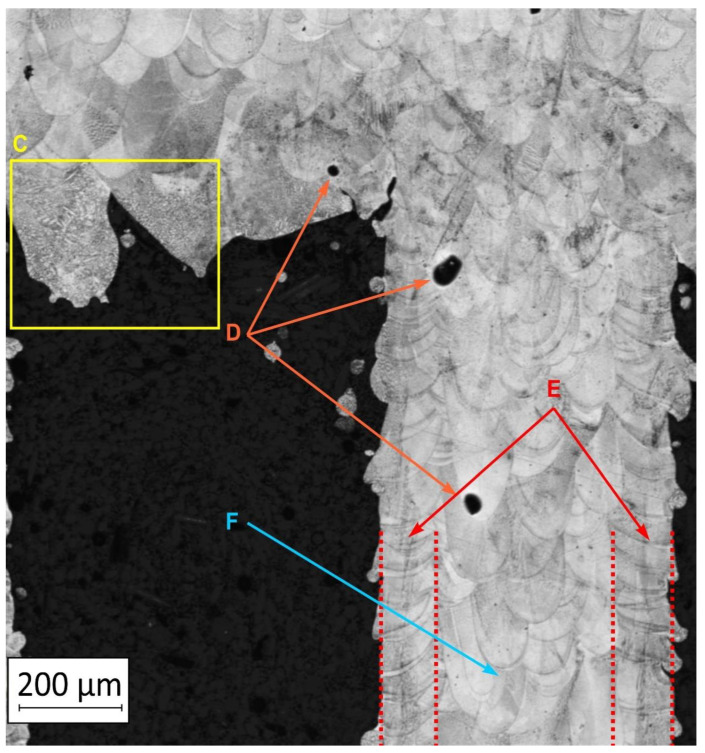
The microstructure of the manufactured sample in the area showing a connection of the thin-walled and monolithic parts with the following elements marked: bridge (**C**), porosity (**D**),the laser-exposure outline shell (**E**), and the laser-exposure core of the part (**F**).

**Figure 5 materials-13-05662-f005:**
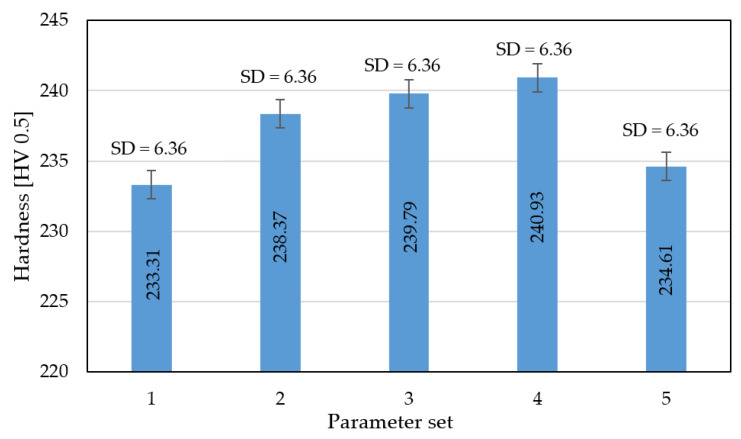
The hardness of parts manufactured using different values of energy density.

**Table 1 materials-13-05662-t001:** Nominal chemical composition of 316L steel.

C	Mn	Si	P	S	N	Cr	Mo	Ni
weight (%)								
max. 0.03	max.	max.	max.	max.	max.	16.00–18.00	2.00–3.00	10.00–14.00
	2.00	0.75	0.04	0.03	0.10			

**Table 2 materials-13-05662-t002:** Physical and mechanical properties of SLM-processed 316L stainless steel.

Type of Parameter	Value
Density ($\frac{g}{cm^{3}})$	7.9
Thermal conductivity ($\frac{W}{mK})$	15
Tensile strength in horizontal orientation (MPa)	651
Tensile strength in vertical orientation (MPa)	640
Offset yield strength in horizontal orientation (MPa)	546
Offset yield strength in vertical orientation (MPa)	529
Elongation at break in horizontal orientation (%)	41
Elongation at break in vertical orientation (%)	43

**Table 3 materials-13-05662-t003:** Sets of analyzed production parameters.

Parameter Set	L_P_ (W)	e_v_ (mm/s)	h_d_ (mm)	l_t_ (mm)	ρ_E_ (J/mm^3^)
1	191	700	0.12	0.05	45.48
2	233	700	0.12	0.05	55.48
3	275	700	0.12	0.05	65.48
4	317	700	0.12	0.05	75.48
5	359	700	0.12	0.05	85.48

**Table 4 materials-13-05662-t004:** Porosity in samples 1–5.

		Parameters Set		
		Value of Registered Porosity (%)	Registered and Merged Image	Porosity of Reference Monolithic Samples (%)
Parameter set	1	0.68	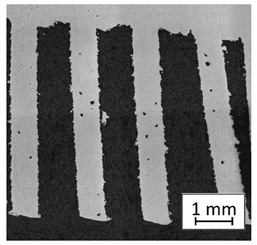	0.59
	2	0.63	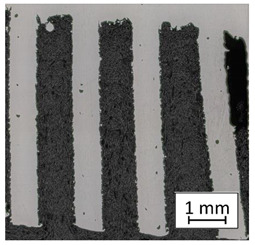	0.12
	3	0.62	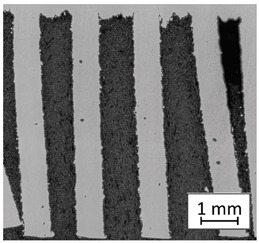	0.10
	4	0.56	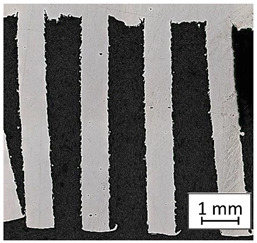	0.03
	5	0.09	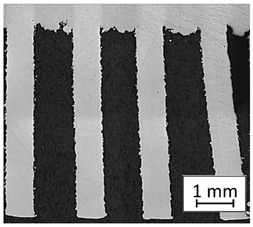	0.04

**Table 5 materials-13-05662-t005:** Surface roughness in samples 1–5.

		Parameters Set	
		Value of Registered Surface Roughness Ra (µm)/Rz (µm)	Registered Roughness Values Map
Parameter set	1	12.56/71.60	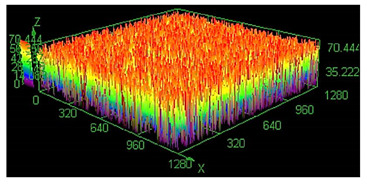
	2	16.36/168.80	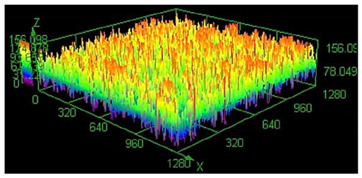
	3	19.54/182.05	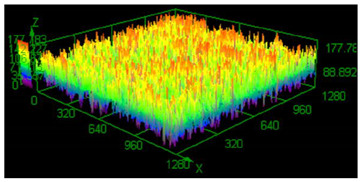
	4	18.43/166.05	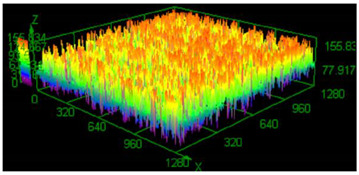
	5	19.71/219.82	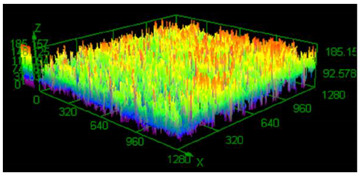

**Table 6 materials-13-05662-t006:** The surface roughness of as-built and surface-treated samples.

	As-Built	Sanded	Electropolished
Surface image	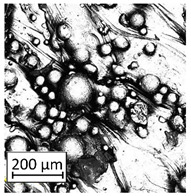	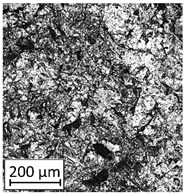	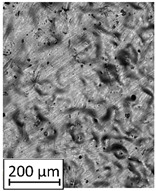
Roughness values map	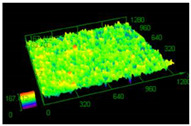	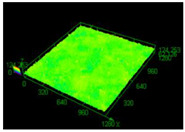	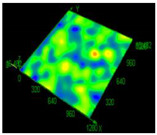
Registered roughness value R_a_ (µm)	13.641	3.950	1.316

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
