# Peer review of "Modification of Structural Properties Using Process Parameters and Surface Treatment of Monolithic and Thin-Walled Parts Obtained by Selective Laser Melting"

_materials, 2020, doi:10.3390/ma13245662_

Round 1
Reviewer 1 Report
The paper “Structural properties modification using process parameters and surface treatment of monolithic and thin-walled parts obtained by Selective Laser Melting” evaluates the modification of the operating parameters in the properties of the resulting material. For this, it proposes five different energy densities and analyzes the porosity, roughness and hardness obtained.
It presents an adequate introduction, establishing the approach of the problem and the current situation in this field of research.
The equipment and methodology are explained correctly. The results obtained are clear, and maybe the authors should support them with the previous research of other authors. The conclusions adequately summarize the work presented. The presentation and structure of the paper is correct, so, in my opinion, it can be published with the following revision:
- Line 128: The authors state: “The recommended energy density for monolithic parts made of 316L steel is about 55 J/mm3”. It should be explained where that statement comes from
- Line 153: “As it is well known, pores…”. It would be convenient if references were included to support this comment.
- Line 160: “Basing on the obtained results it could be observed a lower value of average roughness surface profile (Ra) in samples manufactured using a higher value of energy density”. How do you explain that the result for parameter set 2 does not match this conclusion?
- Line 161: “It is also visible the highest value of the lowest and highest points distribution in samples manufactured using the low value of energy density”. This statement is not clear. Perhaps better images than those presented in table 3 (axis values are not readable) could help to understand what the authors mean.
- Line 212: “Energy density increase of 70% caused almost an 85% decrease in porosity and a 15% increase in surface roughness strength. Additionally, manufacturing using parameters with lower energy density had a slight increase in measured hardness.” The effects achieved with the different energy densities seem contradictory. An increase benefits some properties and others, according to this sentence, might not. It would be appropriate for the authors to establish which is the priority objective.
Author Response
Dear Reviewer,
In the beginning, I would like to thank you for taking your time and giving your valuable comments. Regarding your revision, we made proper corrections which were yellow-highlighted in our manuscript. Based on your comments we made corrections as follows:
- “Line 128: The authors state: “The recommended energy density for monolithic parts made of 316L steel is about 55 J/mm3”. It should be explained where that statement comes from”.
- Now the sentence read as follows: “The recommended energy density (by the SLM Solutions) for monolithic parts made of 316L steel is about 55 J/mm3”. As it is well known the energy density for each device could differ, especially when we are taking into account different producers. In our research, we used the SLM 280HL device and this parameter value is dedicated to this exact machine.
- “Line 153: “As it is well known, pores…”. It would be convenient if references were included to support this comment.” – we put proper citations – thank you for your advice. Mentioned references are:
- Stern, F.; Kleinhorst, J.; Tenkamp, J.; Walther, F. Investigation of the anisotropic cyclic damage behavior of selective laser melted AISI 316L stainless steel. Fatigue Fract. Eng. Mater. Struct. 2019, 7, 1–9.
- Zhang, M.; Sun, C.N.; Zhang, X.; Wei, J.; Hardacre, D.; Li, H. Predictive models for fatigue property of laser powder bed fusion stainless steel 316L. Mater. Des. 2018, 145, 42–54.
- Fatemi, A.; Molaei, R.; Phan, N. Multiaxial fatigue of additively manufactured metals: Performance, analysis, and applications. Int. J. Fatigue 2020, 134, 105479.
- “Basing on the obtained results it could be observed a lower value of average roughness surface profile (Ra) in samples manufactured using a higher value of energy density”. How do you explain that the result for a parameter set 2 does not match this conclusion?
- Dear Reviewer, we would like to thank you and apologize to you at the same time – the 1st and 5th sample had been inverted by a mistake. We have checked our results once again and found this issue. This section has been changed as follows: “Basing on the obtained results it could be observed an increased value of average roughness surface profile (Ra) in monolithic samples manufactured using a higher value of energy density. It is also visible the highest value of the lowest and highest points distribution (visible as the same color – the red top part of the 1st parameter set) in samples manufactured using the low value of energy density. The issue of slight increase and surface roughness stabilization near a value of 19 μm could be connected with increased spattering by providing a material high amount of energy. That phenomenon should be taken into consideration during process preparation, where it is necessary to distinguish part porosity, surface roughness, or both to reach desirable strength/fatigue properties.”
- “Line 161: “It is also visible the highest value of the lowest and highest points distribution in samples manufactured using the low value of energy density”. This statement is not clear. Perhaps better images than those presented in table 3 (axis values are not readable) could help to understand what the authors mean”. We tried to manage somehow those images but there were no clear differences. We decided to put the following explanation:
“It is also visible the highest value of the lowest and highest points distribution (visible as the same color – i.e. red top part of the 1st parameter set) in samples manufactured using the low value of energy density.”
- Line 212: “Energy density increase of 70% caused almost an 85% decrease in porosity and a 15% increase in surface roughness strength. Additionally, manufacturing using parameters with lower energy density had a slight increase in measured hardness.” The effects achieved with the different energy densities seem contradictory. An increase benefits some properties and others, according to this sentence, might not. It would be appropriate for the authors to establish which is the priority objective.
Dear Reviewer, if we would have been asked which parameters set will be the best, the first question will be: - for what kind of solution? As could be stated after our analysis: Thin-walled parts have better density after providing a higher amount of energy into the material during SLM processing, but at the same time, we are achieving a decreased value of hardness. The decision of selecting the proper parameters set is still an open question. In our manuscript, we wanted to explain the existence of that phenomenon – especially that there is no such thing as the proverbial golden mean in additive manufacturing. So we tried to highlight that it is very important to consider process properties before additive manufacturing of monolithic and thin-walled parts.
- To summarize, we hope you found our improvements sufficient. Once again thank you for your comments which could be made a significant improvement to our manuscript.
Sincerely,
Authors
Reviewer 2 Report
The paper reports about “Structural properties modification using process parameters and surface treatment of monolithic and thin-walled parts obtained by Selective Laser Melting”. The topic of the paper has a relevant interest for material science and the advanced technology. Reviewed article is very interesting and write at good scientific level. However, the manuscript requires a revision prior to publication.
The following suggestions have to be addressed before publication of the paper:
1.In the subchapter “Material properties analysis”, addition of the physical and mechanical properties of 316L stainless steel, will increase quality of the manuscript.
2.In Figure 2, please add scale. Also, please add the sample dimensions in text. In text, a lack of reference to the markings such as A and B in Figure 2.
2.In Figure 3, also the A and B markings are indicated. Please use various markings without its repetition.
3.In Figure 4, please add values of the standard deviations.
4.In Table 4, please add also values of the Rz surface roughness parameters.
5.In Line 192, please add more information concerning used the methods to improve the surface quality. Please describe shortly processes such as sanded and electropolished, and applied parameters. Each additional process can change the properties material of the SLM manufactured sample.
6.In Abstract, please indicate particular results from analysis. In Keywords, please add the powder material: 316L stainless steel.
Author Response
Dear Reviewer,
In the beginning, I would like to thank you for taking your time and giving your valuable comments. Regarding your revision, we made proper corrections which were green-highlighted in our manuscript. Based on your comments we made corrections as follows:
- In the subchapter “Material properties analysis”, the addition of the physical and mechanical properties of 316L stainless steel, will increase the quality of the manuscript. – we agreed with your comment and put an additional table with the mentioned properties based on the producer’s datasheet:
Material properties of SLM-processed 316L steel (based on SLM Solution’s Data Sheet) using 50μm layer thickness were shown in table 2.
Table 2. Physical and mechanical properties of SLM-processed 316L stainless steel
|
Type of parameter |
Value |
|
Density (g/cm3) |
7.9 |
|
Thermal conductivity (W/mK) |
15 |
|
Tensile strength in horizontal orientation (MPa) |
651 |
|
Tensile strength in a vertical orientation (MPa) |
640 |
|
Offset yield strength in horizontal orientation (MPa) |
546 |
|
Offset yield strength in a vertical orientation (MPa) |
529 |
|
Elongation at break in horizontal orientation (%) |
41 |
|
Elongation at break in a vertical orientation (%) |
43 |
- In Figure 2, please add scale. Also, please add the sample dimensions in the text. In text, a lack of reference to the markings such as A and B in Figure 2.
- We put a proper scale in figure 2 and additional text: “ Their dimensions were as follows: 100mm length, 10mm width and 12 mm height. Thin-walled parts (area “A” in figure 2) were characterized by a thickness of 0.75mm and 1mm spacing. Monolithic part of the samples in the middle (area “B” in figure 2) had dimensions of 10mm length, 10mm width and 12 mm height.”
- In Figure 3, also the A and B markings are indicated. Please use various markings without its repetition.
- We modified markings to avoid repetition. Now it has the following form: “Considering the obtained microstructure there could be visible non-homogenous and irregular structure of the material in the area of the bridge (area C in figure 3), in the connection between the outline shell (E in figure 3) and the core of the part (F in figure 3) it was observed some pores (D).”
- In Figure 4, please add values of the standard deviations.
- Values of standard deviations have been provided. Additionally, based on your comment, we decided to put exact hardness values on each bar.
- In Table 4, please add also values of the Rz surface roughness parameters.
- Rz values have been put into the table
- In Line 192, please add more information concerning used the methods to improve the surface quality. Please describe shortly processes such as sanded and electropolished, and applied parameters. Each additional process can change the properties material of the SLM manufactured sample.
Dear Reviewer, we put sanding and electropolishing parameters directly after figure 2:
“To reduce surface roughness, it has been also used PK 1200E sandblaster with silica sand (Virgo, Siedlce, Poland). For the additional surface treatment of the AM parts, Struers Lectropol (Struers, Copenhagen, Denmark) was used for electropolishing. For surface-treated samples, each surface was electropolished three times in A3 Struers electrolyte using 35V voltage, 5A amperage, and 40s polishing time.”
We agree with your statement about changing mechanical properties after different surface treatment processes, especially in the top layer where residual stresses are significantly changed. For that kind of analysis, we had to use the sin2ψ method which is unavailable for us right now.
- In the Abstract, please indicate particular results from analysis. In Keywords, please add the powder material: 316L stainless steel.
We put the mentioned keyword and additional part at the end of our abstract: “It has been observed 70% energy density increase caused almost an 85% porosity decrease and a 15% increase in surface roughness. Additional surface treatment of the material allowed for 70-90% roughness reduction.”
- To summarize, we hope you found our improvements sufficient. Once again thank you for your comments which could be made a significant improvement to our manuscript.
Reviewer 3 Report
This paper focused on the effect of laser power on the microstructure and mechanical properties of thin-walled parts fabricated by SLM. The porosity, surface roughness and microhardness were demonstrated. The effects of different surface treatments on surface roughness were also studied. However, the research in this area was innovative. Take the whole article into consideration and give comments on rejection. While if you revise it carefully, it is still hope to be received on this topic. The specific suggestions are as follows:
- It is recommended to supply the powder's microscopic morphology and particle size distribution diagram to facilitate understanding of the actual state of the powder.
- “Indeed, a value of porosity at a level lower than 1% is acceptable”, it is recommended to quote references.
- “it is worth to increase the energy density by about 30% to assure appropriate melted material around the thin-walled structure.” Proposal to supply the basis of this data (30%).
- It is recommended to supply the porosity data of the monolithic in Table 2 for comparison.
- In “Porosity analysis” section, it is recommended to introduce theories, such as heat transfer gradient and heat transfer method related theories to explain the phenomenon.
- The surface roughness measurement section did not indicate the specific method, such as measuring several lines and taking the average value.
- In “Surface roughness analysis” section, roughness and material strength are not necessarily related in theory, please correct.
- This article did not give data or pictures of the monolithic where appropriate.
- The surface roughness data does not have a good linear relationship with the laser power, and no explanation was given in the article
- It is recommended to make the pictures more beautiful.
Author Response
Dear Reviewer,
In the beginning, I would like to thank you for taking your time and giving your valuable comments. Regarding your revision, we made proper corrections which were blue-highlighted in our manuscript. Based on your comments we made corrections as follows:
- “It is recommended to supply the powder's microscopic morphology and particle size distribution diagram to facilitate understanding of the actual state of the powder.”
- We put powder SEM analysis (have a look at figure 4) with powder particle size distribution cumulated mass values.
- “Indeed, a value of porosity at a level lower than 1% is acceptable”, it is recommended to quote references.
- We put proper citations, where authors achieved even 5% porosity and treat it as acceptable.
Jaskari, M.; Mäkikangas, J.; Järvenpää, A.; Mäntyjärvi, K.; Karjalainen, P. Effect of High Porosity on Bending Fatigue Properties of 3D Printed AISI 316L Steel. Procedia Manuf. 2019, 36, 33–41.
Cherry, J.A.; Davies, H.M.; Mehmood, S.; Lavery, N.P.; Brown, S.G.R.; Sienz, J. Investigation into the effect of process parameters on microstructural and physical properties of 316L stainless steel parts by selective laser melting. Int. J. Adv. Manuf. Technol. 2014, 76, 869–879.
- “It is worth to increase the energy density by about 30% to assure appropriate melted material around the thin-walled structure.” Proposal to supply the basis of this data (30%).
The proper citation was included: 25. Tucho, W.M.; Lysne, V.H.; Austbø, H.; Sjolyst-Kverneland, A.; Hansen, V. Investigation of effects of process parameters on microstructure and hardness of SLM manufactured SS316L. J. Alloys Compd. 2018, 740, 910–925.
- It is recommended to supply the porosity data of the monolithic in Table 2 for comparison.
We improved the mentioned table by obtained results for monolithic parts – an added column has been blue-highlighted
- In the “Porosity analysis” section, it is recommended to introduce theories, such as heat transfer gradient and heat transfer method related theories to explain the phenomenon.
We decided to write a dozen-long description of molten pool thermodynamics basics in the interview based on three references:
- Gusarov, A. V.; Yadroitsev, I.; Bertrand, P.; Smurov, I. Heat transfer modeling and stability analysis of selective laser melting. Appl. Surf. Sci. 2007, 254, 975–979.
- Liu, Y.; Yang, Y.; Mai, S.; Wang, D.; Song, C. Investigation into spatter behavior during selective laser melting of AISI 316L stainless steel powder. Mater. Des. 2015, 87, 797–806.
- Matthews, M.J.; Guss, G.; Khairallah, S.A.; Rubenchik, A.M.; Depond, P.J.; King, W.E. Denudation of metal powder layers in laser powder bed fusion processes. Acta Mater. 2016, 114, 33–42.
To explain heat transfer theories properly we have to provide a few-page thermodynamically-based description of this phenomenon, but our manuscript is more focused on material properties than on thermodynamics. We hope you find our improvement in this topis as sufficient.
- The surface roughness measurement section did not indicate the specific method, such as measuring several lines and taking the average value.
We used 3D laser measurement so there was no need to make several measurements.
- In “Surface roughness analysis” section, roughness and material strength are not necessarily related in theory, please correct.
We removed the word “strength” to follow your advice.
- “This article did not give data or pictures of the monolithic where appropriate.”
We put data about monolithic samples porosity in table 4 – it has been blue-highlighted. We decided to not put images of monolithic parts because there was no important phenomenon in pores distribution in monolithic parts.
- “The surface roughness data does not have a good linear relationship with the laser power, and no explanation was given in the article”
Please have a look at table 3 – the only thing we changed was laser power which was the only one parameter which influenced energy density – in that case, we decided to make a relationship between energy density and surface roughness – as it has been done in i.e
Wang, D.; Liu, Y.; Yang, Y.; Xiao, D. Theoretical and experimental study on surface roughness of 316L stainless steel metal parts obtained through selective laser melting. Rapid Prototyp. J. 2016, 22, 706–716.
- “It is recommended to make the pictures more beautiful.”
We put 300 dpi resolution photos and worked in contrast to improve them.
- To summarize, we hope you found our improvements sufficient. Once again thank you for your comments which could be made a significant improvement to our manuscript.
Round 2
Reviewer 3 Report
Most of the problems have been modified, but there are still some minor problems, it is suggested to modify.
1.There is a sentence in Abstract that “It has been observed 70% energy density increase caused almost an 85% porosity decrease and a 15% increase in surface roughness”. There is a data problem in this sentence. “85% porosity decrease” corresponding to 0.63 and 0.09, it corresponding to Parameters 2 and Parameters 5. But in Table3, energy density are 55.48J/mm3 and 85.48J/mm3 respectively. They are not the relationship of “70% energy density increase”.
2. From parameters 1 to Parameters 5, the energy density is monotonically rising, but Surface roughness data 12.56, 16.36, 19.54, 18.43, 19.71. They are not linear relationship. It didn’t show up the data that “15% increase in surface roughness”, which sentence in Abstract and Conclusion.
Author Response
Dear Reviewer,
Regarding your comments:
Ad.1. You are 100% right. We changed properly abstract and also the conclusions part - it has been yellow highlighted.
Ad.2. As you can see a roughness increased and stabilized on some level. It has been described and covered by proper citation - please have a look at line 197. It has been also yellow-highlighted.
We hope you find our corrections and explanations enough to allow publishing our manuscript.
Yours sincerely,
Authors
